# Accuracy of serological tests for diagnosis of chronic pulmonary aspergillosis: A systematic review and meta-analysis

Cláudia Elizabeth Volpe Chaves [1,2]☯*, Sandra Maria do Valle Leone de Oliveira[1]☯, James Venturini[1], Antonio Jose Grande[3], Tatiane Fernanda Sylvestre[4], Rinaldo Poncio Mendes[1,4], Anamaria Mello Miranda Paniago[1]

1 Graduate Program in Infectious and Parasitic Diseases of Federal University of Mato Grosso do Sul, Campo Grande, Mato Grosso do Sul, Brazil, 2 Regional Hospital of Mato Grosso do Sul, Campo Grande, Mato Grosso do Sul, Brazil, 3 State University of Mato Grosso do Sul, Campo Grande, Mato Grosso do Sul, Brazil, 4 Tropical Diseases Department, Faculdade de Medicina de Botucatu, Universidade Estadual Paulista (UNESP), Botucatu, São Paulo State, Brazil

☯ These authors contributed equally to this work.
* claudiavolpe70@hotmail.com

**Data Availability Statement:** All relevant data are within the paper and its Supporting Information files.

## Abstract

Chronic pulmonary aspergillosis (CPA) is a slow and progressive disease that develops in preexisting lung cavities of patients with tuberculosis sequelae, and it is associated with a high mortality rate. Serological tests such as double agar gel immunodiffusion test (DID) or counterimmunoelectrophoresis (CIE) test have been routinely used for CPA diagnosis in the absence of positive cultures. However, these tests have been replaced with enzyme-linked immunoassay (ELISA) and, a variety of methods. This systematic review compares ELISA accuracy to reference test (DID and/or CIE) accuracy in CPA diagnosis. It was conducted according to the Preferred Reporting Items for Systematic Reviews and Meta-Analyses (PRISMA). The study was registered in PROSPERO under the registration number CRD42016046057. We searched the electronic databases MEDLINE (PubMed), EMBASE (Elsevier), LILACS (VHL), Cochrane library, and ISI Web of Science. Gray literature was researched using Google Scholar and conference abstracts. We included articles with patients or serum samples from patients with CPA who underwent two serological tests: ELISA (index test) and IDD and/or CIE (reference test). We used the test accuracy as a result. Original articles were considered without a restriction of date or language. The pooled sensitivity, specificity, and summary receiver operating characteristic curves were estimated. We included 14 studies in the review, but only four were included in the meta-analysis. The pooled sensitivities and specificities were 0.93 and 0.97 for the ELISA test. These values were 0.64 and 0.99 for the reference test (DID and/or CIE). Analyses of summary receiver operating characteristic curves yielded 0.99 for ELISA and 0.99 for the reference test (DID and/or CIE). Our meta-analysis suggests that the diagnostic accuracy of ELISA is greater than the reference tests (DID and/or CIE) for early CPA detection.

**Funding:** This study was financed in part by the Coordination for the Improvement of Higher Education Personnel - Brazil (CAPES) - Finance Code 001 and National Council for Scientific and Technological Development (CNPQ) - grant number 103078 / 2018-5.

**Competing interests:** The authors have declared that no competing interests exist.

## Introduction

Chronic pulmonary aspergillosis (CPA) is a slow and progressive lung disease caused by *Aspergillus* spp. that develops in preexisting cavities in patients with chronic respiratory diseases. Pulmonary tuberculosis is its main predisposing factor, and it has a global prevalence estimated at 1.2 million cases [1]. The prognosis is poor, with 38–85% mortality in 5 years [1, 2].

CPA presents with five clinical forms: aspergillus nodule; pulmonary simple aspergilloma; chronic cavitary pulmonary aspergillosis (CCPA), also called complex aspergilloma; chronic fibrosing pulmonary aspergillosis (CFPA); and subacute invasive pulmonary aspergillosis (SAIA) [3]. Aspergilloma is present in only one-third of patients with CPA [1, 4].

CPA diagnosis is based on suggestive imaging evidence, preferably CT scan, of microbiological infection by *Aspergillus* or an immune response to the agent, maintained for at least 3 months [3].

Serologic tests are indispensable for diagnosis in the absence of positive cultures, and they are considered the best noninvasive tests for diagnosis [5, 6]. These tests may be over 90% positive with precipitins or in the detection of *Aspergillus* IgG [2, 3].

In patients presenting *Aspergillus* in the respiratory tract, the detection of specific serum antibodies differentiates infection from colonization, with a positive predictive value of 100% for infection identification [7]. Initially, antibodies against *Aspergillus fumigatus* were determined by detecting precipitins using double immunodiffusion test (DID) or counterimmunoelectrophoresis (CIE) technique [4, 8, 9] with a sensitivity of 89.3% [5] and a specificity of 100% [10]. These techniques require much time, intense work, and relatively large *A. fumigatus* and patient serum extracts, and they only yield semiquantitative results [6].

The *Aspergillus* IgG antibody test is strongly recommended by the Infectious Diseases Society of America (IDSA) [11]. In practice, precipitation techniques have already been replaced by *Aspergillus* enzyme-linked immunosorbent assay (ELISA) IgG antibody detection test [12]. This is the fastest and most sensitive test [13], producing quantitative results with less *A. fumigatus* extract and patient serum per test, and it is easily automated [6].

Despite its importance, serology for *Aspergillus* IgG detection using ELISA still cannot reach a definitive conclusion on diagnostic performance in CPA; significant differences in sensitivity, specificity, and coefficient of variation need to be explored in cohorts of well-characterized patients [3].

It is very difficult to compare results from in-house IgG ELISA tests between laboratories because of the use of non-standard *A. fumigatus* preparations, and the results are obtained in various quantitative units that are also chosen without standardization. For this reason, commercial tests with standardized preparations and concentrations are being used [6]. Currently, we have commercial tests like ELISA plates for *Aspergillus*-specific IgG antibodies produced by Serion (Germany), IBL (Germany/USA), Dynamiker/Bio-Enoche (China), Bio-Rad (France), Bordier (Switzerland), and Omega/Genesis (UK). We also have specific *Aspergillus* IgG automated systems like Immunolite-Siemens (Germany) and ImmunoCAP (Thermo Fisher Scientific/Phadia), which are fluoroenzyme immunoassay ELISA variants. The main limitation of these tests is that they can only detect antibodies against *A. fumigatus*. In some countries such as India and Japan, 40% of patients with CPA are infected with non-fumigatus strains [2].

Considering the various methods for detecting *Aspergillus* antibodies, use of precipitation tests owing to their low cost, and absence of more precise options for serological diagnosis of CPA, this review of CPA serological diagnosis compared the performance of the precipitation tests with enzyme-linked immunoassay tests.

## Materials and methods

We conducted a systematic literature review in accordance with the recommendations of the Preferred Reporting Items for Systematic Reviews and Meta-Analyses (PRISMA) [14] and

STARD 2015 [15]. A systematic review protocol was developed and registered in the International Prospective Register of Systematic Reviews—CRD42016046057. We used the Cochrane recommendations to report systematic reviews and meta-analyses of studies on diagnostic accuracy [16].

### Eligibility criteria

The inclusion criteria comprised studies in which population or serum samples from patients diagnosed with aspergilloma or CPA were subjected to immunoenzymatic test (ELISA) and to DID and/or CIE test. The accuracy of the tests was defined as the primary outcome. Original studies were included without restriction based on language, geographical location, or publication date. We excluded studies with children or animals and *in vitro* studies. We could not find an article in Japanese, which was selected for full article reading because it was not available in the international library commuting service.

### Information sources and search strategies

The following databases were searched for studies: MEDLINE (through PubMed), EMBASE (through Elsevier), LILACS (through VHL), Cochrane library, and ISI Web of Science. Gray literature was researched in Google Scholar and congress abstracts. We performed the search strategy until June 2019.

We used the following search strategy for MEDLINE and adapted it for the other databases: pulmonary aspergillosis AND serologic test (and its synonyms). ("Pulmonary Aspergillosis" [Mesh] or Aspergillosis, Pulmonary or Pulmonary Aspergillosis or Lung Aspergillosis or Aspergillosis, Lung or Aspergillosis, Lung or Bronchopulmonary Aspergillosis or Aspergillosis, Bronchopulmonary or Bronchopulmonary Aspergillosis or Aspergillosis, Bronchopulmonary or Aspergillose, Bronchopulmonary or Bronchopulmonary Aspergillose) AND ("Serologic Tests" [Mesh] or Serological Tests or Serological Tests or Serological Tests, Serological or Tests, Serologic or Serologic Tests or Serologic Tests or Serodiagnoses).

### Study selection and data extraction

Titles were imported from EndNote Online, and duplicate studies were removed. The remaining titles were independently reviewed by two authors (TFS and SMVLO), who selected the article abstracts and summarized the complete texts for evaluation. The divergences were resolved by a third expert reviewer (RPM). Two other authors (CEVC and JV) performed independent evaluations of the complete articles and judged the methodological quality of the included studies using the Quality Assessment of Diagnostic Accuracy Studies (QUADAS-2) tool [17]. The divergences were resolved by consensus among the researchers.

- Two reviewers (CEVC, JV) independently extracted the following data from each included study:

- Study characteristics: author, year of publication, country, design, and sample size;

- Population characteristics: according to the inclusion criteria;

- Description of the index test and cut-off points;

- Description of the reference standard and cut-off points;

- QUADAS-2 items; and

- Accuracy results obtained in each study to construct a diagnostic contingency (2 × 2 table).

## Assessment of methodological quality

For this review, we used the QUADAS-2 tool to assess the methodological quality of studies [17]. QUADAS-2 consists of four key domains: patient selection, index test, reference standard, and flow and timing. We assessed all domains for risk of bias (ROB) potential and the first three domains for applicability concerns. Risk of bias was judged as "low," "high," or "unclear." Two review authors independently completed QUADAS-2 and resolved disagreements through discussion.

## Statistical analysis and data synthesis

We used data reported in the true positive (TP), false positive (FP), true negative (TN), and false negative (FN) format to calculate sensitivity and specificity estimates and 95% confidence intervals (CIs) for individual studies. Summary positive (LR+) and negative (LR-) likelihood ratios and summary diagnostic odds ratios (DOR) were obtained from bivariate analysis. We used the clinical interpretation of likelihood ratios [18] as follows: conclusive evidence (LR+ > 10 and LR- < 0.1), strong diagnostic evidence (LR+ > 5 to 10 and LR- 0.1 to < 0.2), weak diagnostic evidence (LR+ > 2 to 5 and LR- 0.2 to < 0.5), and negligible evidence (LR+ 1–2 and LR- 0.5–1).

In studies where it was possible to calculate sensitivity and specificity for the ELISA test and DID and/or CIE, we calculated the accuracy test and Youden's J statistic. Youden's index values range from zero to one inclusive, with the expectancy that the test will show a greater proportion of positive results for the diseased group than the control [19].

Studies were submitted to meta-analysis when three conditions were met: sample size greater than 20; sensitivity and specificity were available for the index and the reference tests; and control group was included in the analysis. We presented individual studies and pooled results graphically by plotting sensitivity and specificity (and their 95% CIs), heterogeneity, and receiver operating characteristic (ROC) space estimates using Stata software. For the subgroup analysis, we presented individual studies and pooled results in forest plots using Meta-DiSc software.

## Investigations of heterogeneity

We investigated heterogeneity using subgroup analysis. First, we analyzed a subgroup with three studies that presented only healthy controls, maintaining high heterogeneity. Next, we analyzed a second subgroup with two of the most recent commercial testing studies. Thus, we found the main source of heterogeneity: in-house and commercial tests. In-house tests present many technical differences. We considered an I2 value close to 0% as having no heterogeneity between studies; close to 25%, low heterogeneity; close to 50%, moderate heterogeneity; and close to 75%, high heterogeneity between studies [20].

# Results

## Study inclusion

A total of 2160 articles were identified. Among these, 2096 were found using a database, and 64 were identified from other sources (manual search). After removing duplicates, 1797 articles remained. After title/abstract exclusion, only 21 articles were submitted to a full text read, and 14 of these were included for the systematic review. Only four studies were included in the meta-analysis (see Fig 1).

## Characteristics of the studies

The characteristics of the included studies are presented in S1 Table. The earliest study was published in 1983 [21], and the five most current articles were published in 2015 [22], 2016

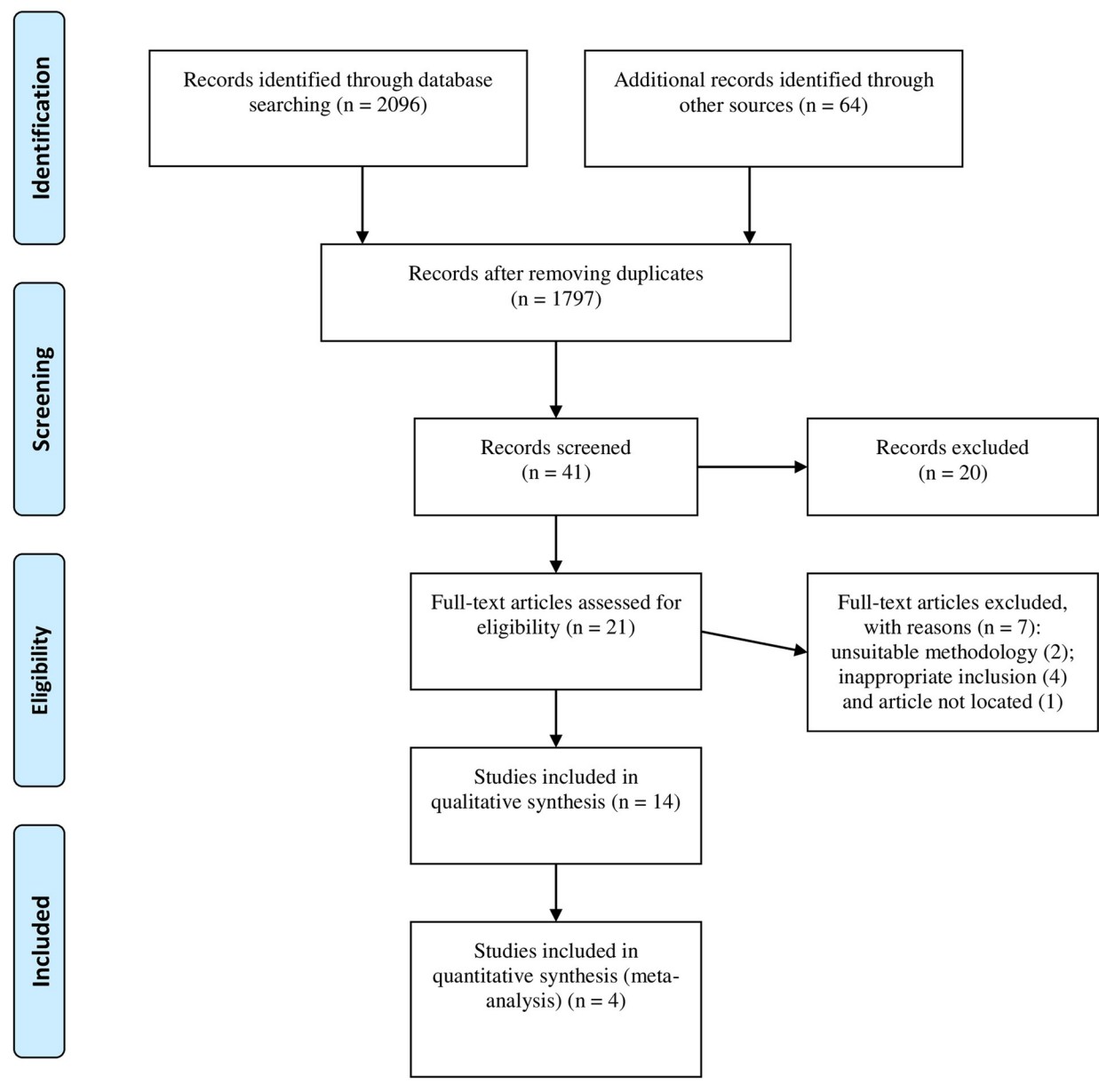

**Fig 1. Study flow diagram.**

[23–25], 2018 [26] and 2019 [27]. Ten studies took place in five countries: Japan [24, 28], Brazil [22], United Kingdom [23, 27, 29], France [25, 30, 31], and India [26]. The study countries were not reported in four articles [21, 32–34].

Ten articles presented DID as the reference test [21, 22, 24, 26–28, 30, 32–34]. One article presented two reference tests, DID and CIE [34]. Four studies presented only CIE as the reference test [23, 25, 29, 31].

Some important differences that were observed after data extraction are highlighted. Seven studies were conducted using in-house ELISA tests [21, 22, 28, 30, 32–34], and seven using commercial tests [23–27, 29, 31]. For both in-house ELISA and commercial tests, different *Aspergillus* antigens and cut-off points were used, beyond those established by the manufacturer (S1 Table).

In one article, we could not identify the number of patients with CPA that was evaluated nor was it possible to extract data from the 2 × 2 table for DID and ELISA [28]. In two articles, it was not possible to recover the DID data [24, 30]. In another article, data were not obtained from CIE [31]. In yet another article [32], it was not possible to extract ELISA data. In one study [33], 20 sera from 13 patients were used; it was not possible to extract the accurate data per patient, and control group data were not presented for the ELISA test. In three articles, the tests did not include a control group [25, 27, 29]. In one article, the control group included patient samples showing the presence of DID precipitation lines; we did not consider this to be a control group [24]. Only one study used participants with disease as controls [26].

During the extraction of ELISA antigen concentration data, five studies using in-house tests presented concentrations varying from 0.1 mcg to 250 mcg per well [21, 22, 30, 33, 34]. These concentrations were not reported in two other articles [28, 32].

Additional differences were found between the in-house tests in studies, including ELISA secondary antibody dilution, with concentrations ranging from 1:100 to 1:300, when described [21, 22, 33, 34]. These dilutions were not reported in three articles [28, 30, 32]. When we evaluated the cut-off for ELISA, several descriptions were found with titers ranging from 1:100 to 1:800. We also found values in OD (optic density), AU/mL, in percentage, and in absorbance. There was no comparable value in in-house tests [21, 22, 33, 34]. The cut-off was not described in three articles [28, 30, 32]. For the ELISA substrate, TMB (3,3′,5,5′-tetramethylbenzidine) was found in two articles [21, 22], pNPP (alkaline phosphatase yellow) in two [33, 34], and OPD (*o*-phenylenediamine) in one [30]. The substrate was not reported in two articles [28, 32].

When extracting antigen concentration data from *A. fumigatus* in the studies using DID or CIE, we found variations between 5 mg/mL and 100 mg/mL [21, 29, 32–34]. We found values expressed in microliters in the following studies: 2 μL [31], 10 μL [25], and 20 μL [23]. Different concentrations were used for somatic antigen [20 mg/mL] and antigen filtration [2 mg/mL) in one article [29]. The antigen concentrations data were not described in four articles [22, 27, 28, 30].

The studies with commercial ELISA tests used the following: ImmunoCap [23, 24, 26, 27, 29], Platelia [29], Immulite [23], Serion [23, 31], Dynamiker [23], Genesis [23], Bio-Rad [25, 31], and Bordier [25]. These tests presented different cut-off points, and the one with the best performance is described in S1 Table.

All methodological differences are listed in S1 Table.

## Risk assessment of bias

We illustrated the methodological quality of the 14 included studies using the QUADAS-2 tool (Figs 2 and 3). All studies had unclear or high risk of bias in at least one domain. Almost all studies [21–24, 26–34] demonstrated high-risk patient selection bias, except one that was unclear [25]. This resulted mainly from not using consecutive or randomized patient samples and using a case-control study. There was no clear definition of exclusion criteria in seven studies [21, 28, 30–34].

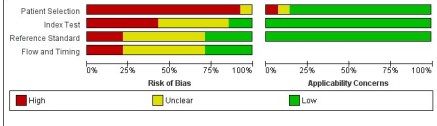

**Fig 2. Proportion graph of studies assessed as having low, high, or unclear risk of bias and/or applicability concerns.**

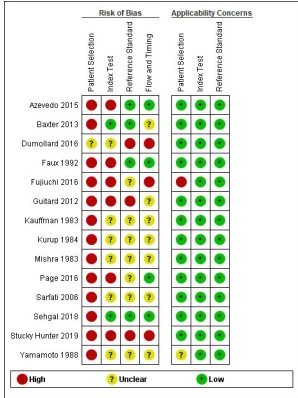

**Fig 3. Risk of bias and applicability concerns graph: Review of the authors' judgments about each domain presented as percentages across included studies.**

In the index test, twelve studies [21–25, 27, 28, 30–34] presented an unclear or high risk of bias. This was mainly because the index test was interpreted with prior knowledge of the standard test. Twelve studies had a low risk of bias in the previous cut-off determination [21–29, 31, 33, 34].

In the reference test, all studies had a low risk of correctly classifying the target condition. Bias risk assessment was uncertain or high risk in ten studies [21, 23–25, 27, 28, 30, 31, 33, 34] owing to a lack of clarity regarding whether the standard test was interpreted without the knowledge of the index test or with prior knowledge.

Regarding flow and time, bias risk assessment was uncertain in nine studies [21, 25, 27–31, 33, 34] for not clearly describing whether there was an appropriate interval between conducting the index test and the reference test. The evaluation was high risk in two studies [24, 27]. All patients were submitted to a reference test in eleven studies, which were included in the analysis [21–24, 26, 29–34]; the results showed low risk. Not all patients were submitted to a test reference in two studies [25, 27], and this was uncertain in one study [28].

Almost all the articles presented low applicability concern, because they did not fail to correspond to the critical question in our study.

## Diagnostic accuracy

We present all articles included in this systematic review with a description of the index and reference tests, the number of patients and control groups, and the values of sensibility, specificity, accuracy test, likelihood positive value, likelihood negative value, and Youden's statistic in Table 1.

The Youden index ranged from 0.50 to 0.98 for the ELISA test and from 0.26 to 1 for the reference test (DID and/or CIE) for the individual studies. Four studies presented a good performance above 0.90 Youden index for the ELISA test [23, 26, 31, 34] and three studies for the reference test [21, 32, 34]. Two studies used commercials tests [23, 26] using the fluorescent enzyme immunoassay method with the ImmunoCAP system, and the best cut-off value for this test in our study (sensitivity: 100%, specificity: 96%) was 27 mgA/L [26]. The other studies presented a performance below 0.90. The Youden index indicates the trade-off between sensitivity and specificity.

## Quantitative synthesis and meta-analysis

In individual studies included in the meta-analysis, ELISA test sensitivity ranged from 0.83 (95% CI 0.63–0.95) [21] to 0.96 (95% CI 0.93–0.98) [23], and specificity ranged from 0.92

**Table 1. Performance of ELISA test and reference tests in studies included in the systematic review.**

| Ref. | Assay | CPA | Cut-off (ELISA) | Control Group | Sensitivity (%) | Specificity (%) | Accuracy | LR+ | LR- | Youden's J statistic |
|---|---|---|---|---|---|---|---|---|---|---|
| Azevedo et al. [22] | ELISA in-house[a] | 22 | 0,120(OD) | 200 | 81.8 | 94 | 93 | 13.64 | 0.193 | 0.76 |
| | ELISA In-house[b] | 22 | 0,130(OD) | 200 | 72.7 | 97 | 95 | 29.09 | 0.280 | 0.7 |
| | ELISA in-house[c] | 22 | 0,090(OD) | 200 | 86.4 | 96.5 | 96 | 24.68 | 0.141 | 0.83 |
| | ELISA In-house[d] | 22 | 0,100(OD) | 200 | 59.1 | 99.5 | 96 | 118.18 | 0.411 | 0.59 |
| | DID 1[e] | 22 | - | 200 | 45.5 | 100 | 95 | 183.52 | 0.545 | 0.46 |
| | DID 2[f] | 22 | - | 200 | 59.1 | 100 | 96 | 235.96 | 0.414 | 0.59 |
| Baxter et al. [29] | ELISA ImmunoCAP | 116 | >40 mg/dL | - | 86 | - | - | - | - | - |
| | ELISA Platellia | 116 | ≥10 AU/mL | - | 85 | - | - | - | - | - |
| | CIE | 116 | - | 0 | 56 | - | - | - | - | - |
| Dumollard et al. [25] | ELISA Bordier | 129 | ≥1 (OD) | 0 | 98 | - | - | - | - | - |
| | ELISA Bio-Rad | 129 | ≥10 AU/mL | 0 | 95 | - | - | - | - | - |
| | CIE | 129 | - | 0 | 87 | | - | - | - | - |
| Faux et al. [32] | ELISA In-house | 11 | - | 18 | - | - | - | - | - | - |
| | DID | 11 | - | 18 | 100 | 100 | 100 | 36.42 | 0.04 | 1 |
| Fujiuchi et al. [24] | ELISA ImmunoCAP | 96[g] | 50 mgA/L | - | 98 | - | - | - | - | - |
| | ELISA ImmunoCAP | 51[h] | 50 mgA/L | - | 39 | - | - | - | - | - |
| | DID | 147 | - | - | - | - | - | - | - | - |
| Guitard et al. [31] | ELISA Serion | 51 | >70 U/mL | 222 | 92/88[t] | 95.9/91[t] | 95/90[t] | - | - | 0.88/0.79[t] |
| | ELISA Bio-Rad | 51 | ≥10 U/mL | 222 | 94/90[u] | 100/99.5[t] | 100/99[t] | - | - | 0.94/0.9[t] |
| | CIE | 51 | - | 222 | - | - | - | - | - | - |
| Kauffman et al. [33] | ELISA In-house | 20 (13)[i] | >31,5%; >1:200 | 50 | - | - | - | - | - | - |
| | DID | 20 (13)[i] | - | 50 | - | - | - | - | - | - |
| Kurup et al. [21] | ELISA in-house[j] | 24 | 1:400 | 12 | 83.3 | 100 | 88.9 | 21.32 | 0.19 | 0.83 |
| | ELISA in-house[k] | 24 | 1:400 | 12 | 50 | 100 | 66.7 | 13.00 | 0.52 | 0.5 |
| | ELISA in-house[l] | 24 | 1:400 | 12 | 79.2 | 100 | 86.1 | 20.28 | 0.23 | 0.79 |
| | DID 507[j] | 24 | - | 12 | 95.8 | 91.7 | 94.4 | 11.50 | 0.05 | 0.88 |
| | DID 534[k] | 24 | - | 12 | 100 | 83.3 | 94.4 | 5.10 | 0.03 | 0.83 |
| | DID 515[l] | 24 | - | 12 | 96 | 100 | 97.2 | 24.44 | 0.06 | 0.96 |
| Mishra et al. [34] | ELISA In-house | 17 | >1:800; 0,3[v] | 50 | 100 | 98 | 98.5 | 33.06 | 0.03 | 0.98 |
| | DID | 17 | - | 50 | 100 | 100 | 100 | 99.17 | 0.03 | 1 |
| | CIE | 17 | - | 50 | 100 | 100 | 100 | 99.17 | 0.03 | 1 |
| Page et al. [23] | ELISA ImmunoCAP | 341 | 20 mg/L | 100 | 96 | 98 | 96 | 47.95 | 0.04 | 0.94 |
| | ELISA Immulite | 341 | 10 mg/L | 100 | 96 | 98 | 96 | 47.95 | 0.04 | 0.94 |
| | ELISA Serion | 341 | 35 U/mL | 100 | 90 | 98 | 92 | 44.87 | 0.11 | 0.88 |
| | ELISA Dynamiker | 341 | 65 AU/mL | 100 | 77 | 97 | 82 | 25.71 | 0.24 | 0.74 |
| | ELISA Genesis | 341 | 20 U/mL | 100 | 75 | 99 | 80 | 75.07 | 0.25 | 0.74 |
| | CIE | 341 | - | 100 | 59 | 100 | 68 | 119.01 | 0.41 | 0.59 |
| Sarfati et al. [30] | ELISA In-house[m] | 51 | - | 41 | 81 | 98 | 88 | 33.09 | 0.20 | 0.79 |

*(Continued)*

**Table 1.** (Continued)

| Ref. | Assay | CPA | Cut-off (ELISA) | Control Group | Sensitivity (%) | Specificity (%) | Accuracy | LR+ | LR- | Youden's J statistic |
|------|-------|-----|-----------------|---------------|-----------------|-----------------|----------|-----|-----|----------------------|
| | ELISA In-house[n] | 51 | - | 41 | 79 | 98 | 87 | 32.37 | 0.22 | 0.77 |
| | ELISA In-house[o] | 51 | - | 41 | 77 | 98 | 86 | 31.65 | 0.23 | 0.75 |
| | ELISA In-house[p] | 51 | - | 41 | 93 | 95 | 94 | 19.06 | 0.07 | 0.88 |
| | ELISA In-house[q] | 51 | - | 41 | 93 | 95 | 94 | 19.06 | 0.07 | 0.88 |
| | ELISA in-house[r] | 51 | - | 41 | 91 | 95 | 93 | 18.70 | 0.09 | 0.86 |
| | ELISA in-house[s] | 51 | - | 41 | 95 | 93 | 94 | 12.95 | 0.06 | 0.88 |
| | DID | 51 | - | 41 | - | - | - | - | - | - |
| Sehgal et al. [26] | ELISA ImmunoCAP | 137 | 27 mgA/L | 50[u] | 94 | 100 | 96 | 95.72 | 0.06 | 0.96 |
| | DID | 137 | - | 50[u] | 26 | 100 | 46 | 26.24 | 0.75 | 0.26 |
| Stucky Hunter et al. [27] | ELISA ImmunoCAP | 154 | 20mgA/L | - | 94 | - | - | - | - | - |
| | ELISA ImmunoCAP | 154 | 40mgA/L | - | 81 | - | - | - | - | - |
| | ELISA ImmunoCAP | 154 | 50mgA/L | - | 71 | - | - | - | - | - |
| | DID | 108 | - | - | 57 | - | - | - | - | - |
| Yamamoto et al. [28] | ELISA in-house | - | - | 45 | - | - | - | - | - | - |
| | DID | - | - | - | - | - | - | - | - | - |

a. AF (*A. fumigatus*) strain

b. AF, *A. niger* and *A. flavus* pool

c. AF strain

d. AF, *A. niger*, and *A. flavus* pool

e. AF strain

f. AF, *A. niger*, and *A. flavus* pool

g. proven cases

h. possible case

i. 20 patients (13 sera)

j. AF 507 strain

k. AF 537 strain

l. AF 515 strain

m. RNU

n. DPPV

o. CAT

p. CAT + RNU

q. CAT+ DPPV

r. RNU + DPPV

s. RNU + DPPV + CAT

t. first and second percentages were obtained then equivocal results were considered as positives or negatives, respectively

u. Diseased controls were used in this study

v. absorbance.

(95% CI 0.64–1.00) [21] to 0.98 (95% CI 0.93–1.00) [23]. The pooled sensitivity and specificity for the ELISA test based on four data studies [21–23, 26] were 0.93 (95% CI 0.87–0.96) and 0.97 (95% CI 0.94–0.98), respectively. Pooled LR+ and LR- were 31.40 (95% CI 16.40–60.10) and 0.07 (95% CI 0.04–0.14), respectively. Pooled DOR was 440.00 (95% CI 156.00–1241.00). We interpreted the pooled LR+/LR- from the ELISA test as conclusive evidence, but we have

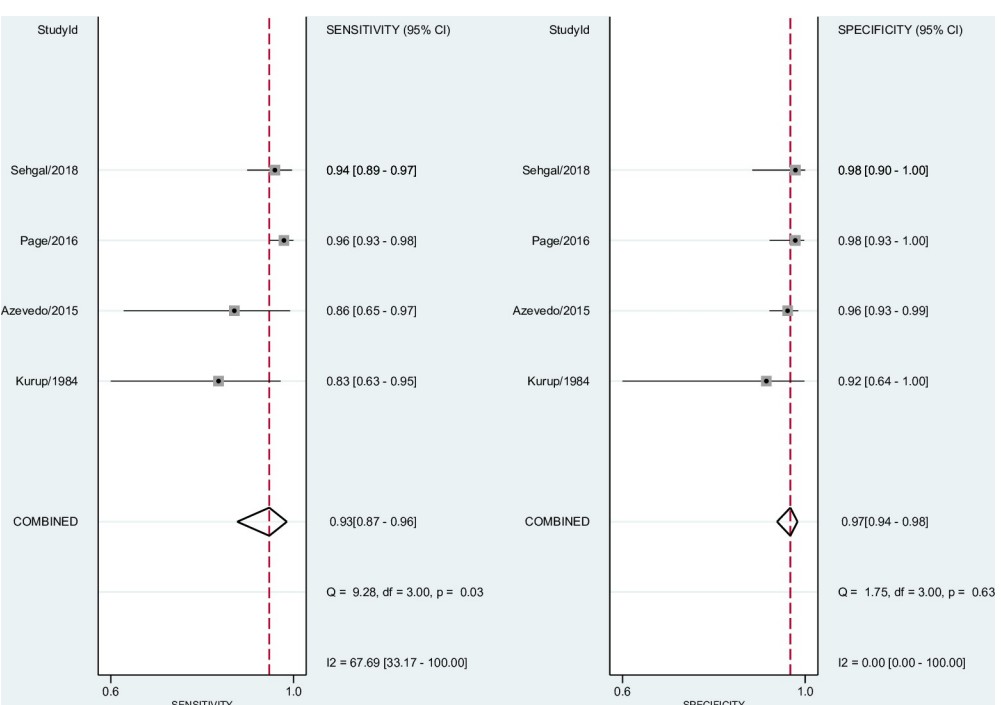

**Fig 4. Forest plot for sensitivity, specificity, and heterogeneity from four ELISA studies.**

not interpreted the reference test (DID and/or CIE) in the same way because LR- was included as weak diagnostic evidence.

In the DID and/or CIE test analyses, the sensitivity and specificity in individual studies ranged from 0.26 (95% CI 0.18–0.34) [26] to 0.96 (95% CI 0.79–1.00) [21] and 0.92 (95% CI 0.64–1.00) [21] to 1.00 (95% CI 0.97–1.00) [22], respectively. The pooled sensitivity and specificity for DID and/or CIE tests were 0.64 (95% CI 0.29–0.89) and 0.99 (95% CI 0.96–1.00). Pooled LR+/LR- were 53.00 (95% CI 19.20–146.40) and 0.36 (95% CI 0.14–0.92). Pooled DOR was 146.00 (95% CI 40.00–532.00).

The forest plots in Figs 4 and 5 show the sensitivity, specificity ranges, and heterogeneity for the ELISA test and reference test (DID and/or CIE) in detecting CPA across the included studies.

We also constructed the sROC curves and calculated the area under the ROC (AUROC) for included studies (Fig 6). The overall diagnostic performance of ELISA and the reference tests (DID and/or CIE) were comparable (AUROC 0.99 [95% CI 0.97–0.99] and 0.99 [95% CI 0.97–0.99], respectively).

## Heterogeneity investigations

When we evaluated the four studies [21–23, 26], we found a heterogeneity (I2) of 67.69 (95% CI 33.17–100.00) in the ELISA sensitivity pool, considered as moderate heterogeneity, and 96.50 (95% CI 94.38–98.62) in the DID and/or CIE sensitivity pool, considered to be highly heterogeneous. First, we investigated the subgroup analyses, evaluating only the three studies using healthy controls [21–23]. We found a heterogeneity (I2) of 72.40% in the ELISA sensitivity pool and 88. 20% in the DID and/or CIE sensitivity pool, considered as high heterogeneity. These results are presented in Figs 7 and 8.

Next, we investigated the second subgroup analyses, evaluating only the two most recent studies using commercial ELISA tests [23, 26]. The heterogeneity (I2) was 0% for sensitivity

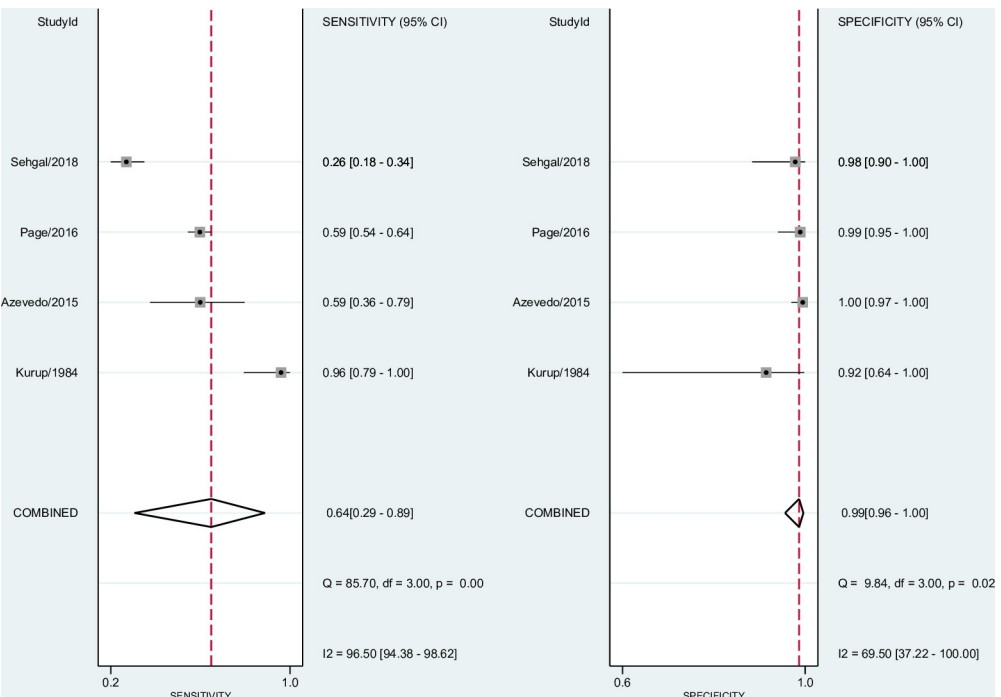

**Fig 5. Forest plot for sensitivity, specificity, and heterogeneity from four DID and/or CIE studies.**

and specificity. When we studied the reference tests, the heterogeneity (I2) was 97.8% for sensitivity and 0% for specificity.

The pooled sensitivity and specificity for the ELISA test based on two data studies [23, 26] were 0.95 (95% CI 0.93–0.97) and 0.98 (95% CI 0.95–1.00), respectively. Pooled LR+ and LR-

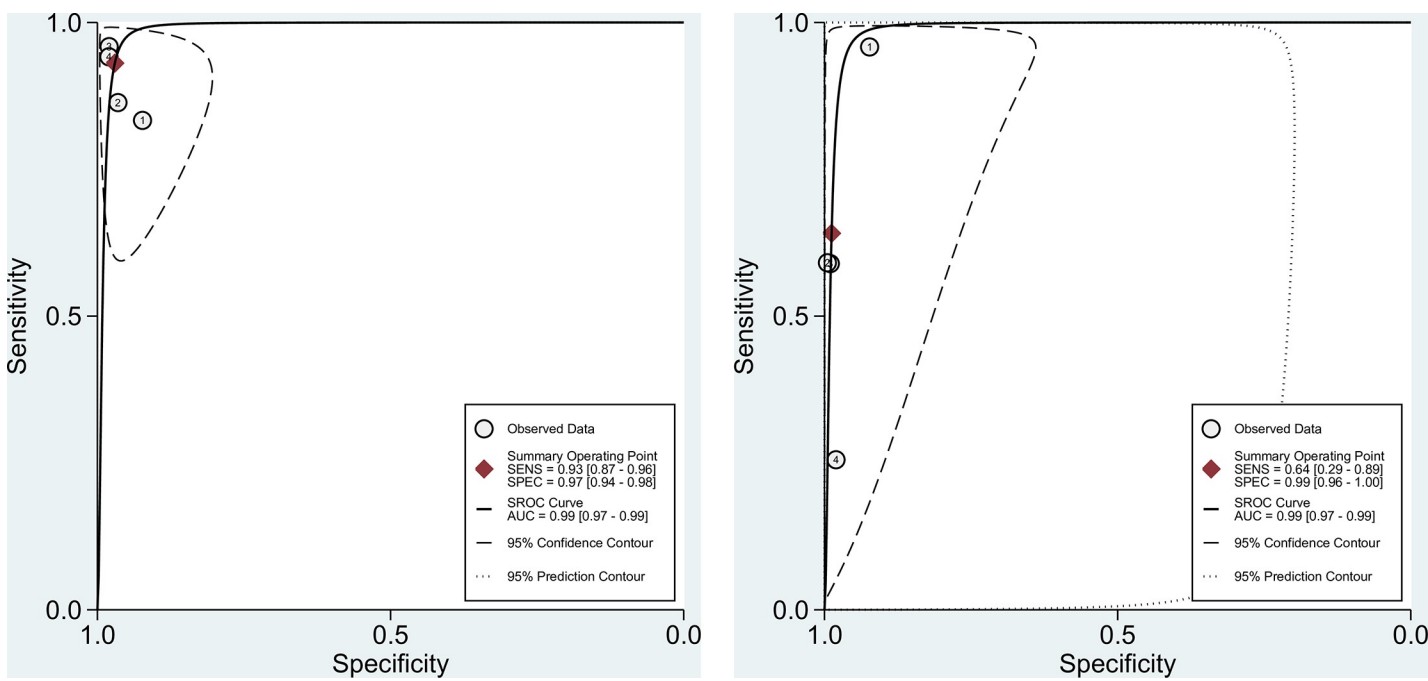

**Fig 6. Summary ROC curves from the four included studies.** A. AUROC for ELISA test; B. AUROC for reference test (DID and/or CIE).

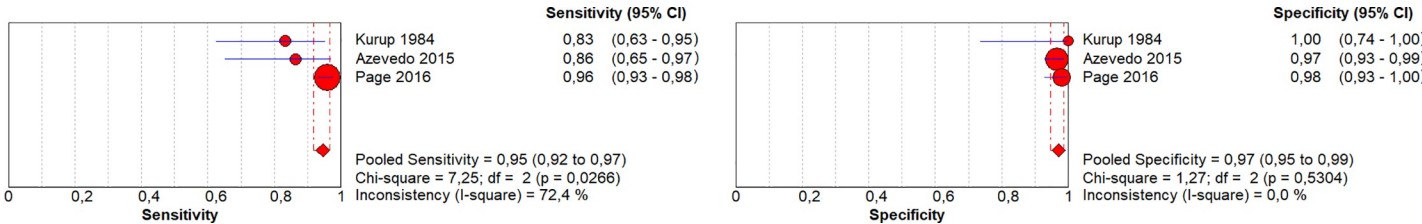

**Fig 7.** Forest plot of sensitivity (A), specificity (B), and heterogeneity from the ELISA test for the subgroup analyses (three studies with healthy controls).

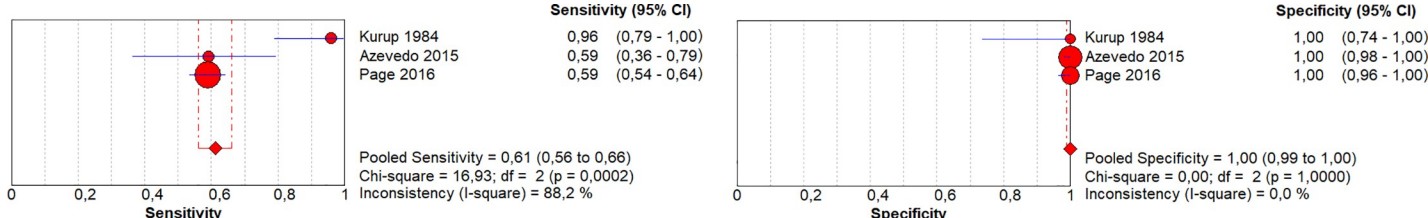

**Fig 8.** Forest plot of sensitivity (A), specificity (B), and heterogeneity from the DID and/or CIE test for the subgroup analyses (three studies with healthy controls).

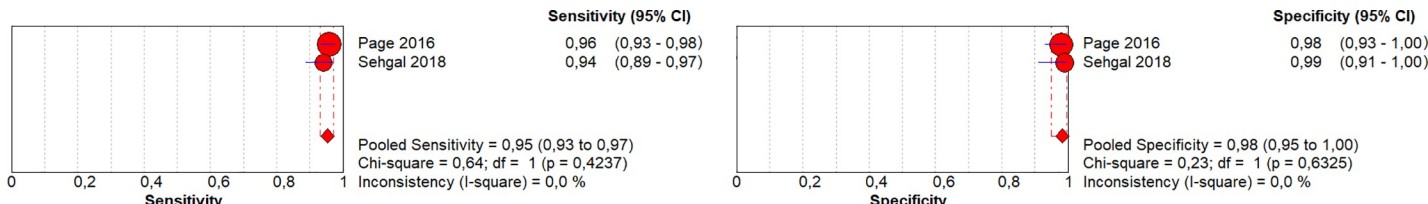

**Fig 9.** Forest plot of sensitivity (A), specificity (B), and heterogeneity from the ELISA test for the subgroup analyses (two studies with commercial tests).

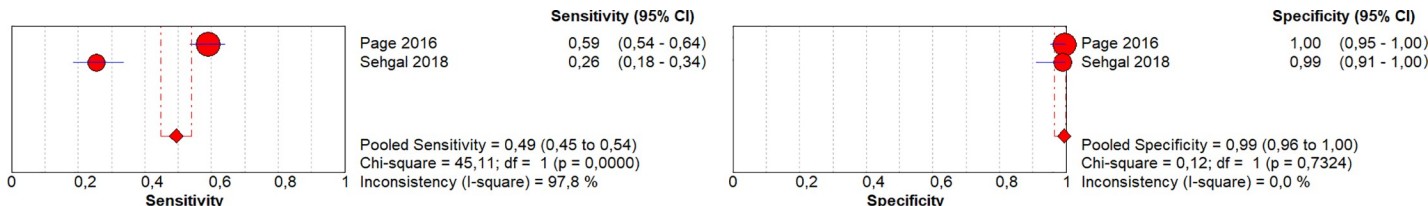

**Fig 10.** Forest plot of sensitivity (A), specificity (B), and heterogeneity from the DID and/or CIE test for the subgroup analyses (two studies with commercial tests).

were 54.92 (95% CI 16.08–187.64) and 0.05 (95% CI 0.03–0.07), respectively. Pooled DOR was 1231.40 (95% CI 326.00–4651.70). The pooled sensitivity and specificity for the reference test (DID and/or CIE) based on two data studies [23, 26] were 0.49 (95% CI 0.45–0.54) and 0.99 (95% CI 0.96–1.00), respectively. Pooled LR+ and LR- were 55.39 (95% CI 7.82–392.60) and 0.56 (95% CI 0.29–1.06), respectively. Pooled DOR was 100.07 (95% CI 11.84–845.84). These results are presented in Figs 9 and 10.

Studies using in-house ELISA tests show large methodological differences in their performance. High heterogeneity was maintained for sensitivity in both studies using DID and/or

CIE tests [23, 26], considering that the precipitation tests are all in-house and also present large methodological differences in the studies included in this review.

## Discussion

To our knowledge, this is the first systematic review to compare ELISA test with precipitin tests (DID and/or CIE) for CPA diagnosis. Although current studies suggest ELISA as a better performing test for CPA diagnosis, precipitation tests are still considered to be the reference test in many countries, especially in Brazil where this review was performed.

Fourteen articles that met the criteria for the research question were included, and all studies were considered to have an uncertain or high risk of bias in some domains in the quality risk assessment.

Important methodological differences were verified, mainly related to the in-house ELISA tests. More recent studies with commercial ELISA tests were included in the review, with the differences described. We also observed this phenomenon in DID and/or CIE tests, because these are all still in-house.

We observed mainly in the former studies that population selection was based on stored samples from patients already diagnosed with CPA and submitted to tests described in the review. In addition, the lack of a checklist in the study descriptions was evident. Many QUA-DAS-2 items were not clearly reported, interfering with the quality evaluation. As an example, although we were skilled in extracting the data for constructing the $2 \times 2$ table, we noted that the discussion and conclusion of one study had an error in printing that was not compatible with the objective, methods, and results of the article [21].

The best performances in the ELISA evaluation of individual studies included in the meta-analysis based on the Youden's test were from the commercial tests [23, 26], ImmunoCAP and Immulite tests, which ranged from 0.94 to 0.96. The best cut-off from the ImmunoCAP system in the individual studies was 27 mgA/L [26].

Our study shows several methods used to identify *A. fumigatus*-specific IgG. For in-house testing, we observed a variety of concentrations and antigens used. For commercial tests, there is also no standard cut-off values. The CPA category could justify different values and the possibility of other etiologies causing fungus ball [35]. Other possibilities for different cut-off values observed in our study may be related to the use of healthy or disease controls and ethnic differences.

When we evaluated Youden's J statistic for the precipitation test (DID or CIE) in the studies included in the meta-analysis, only one study presented a performance of 0.96 [21]. The performance for the other studies [22, 23, 26] ranged between 0.26 and 0.59.

The limitations regarding the use of the precipitin test are based on the requirement for immunodiffusion and electrophoresis migration methods. They do not present antigen standardization, besides requiring additional work and much time to obtain the results, especially in low resource countries [36].

The ELISA test seems to be promising. However, even with important methodological differences, it was useful to evaluate the use of diagnostic data for CPA in each study where it was possible to obtain data for sensitivity and specificity calculation. Two more recent studies were highlighted in this review [23, 26], with sensitivities presenting low confidence intervals for the ELISA test. These studies showed a better performance than the confidence intervals from the reference tests (DID and/or CIE). Besides that, the pooled LR+/LR- from the ELISA test presented conclusive evidence, and this was not observed in the reference test results.

Several studies have recently been published with serological data using only commercial ELISA tests for CPA diagnosis in an area with high tuberculosis prevalence [1, 12, 37].

The limitations of this study depend on the primary studies. There were problems regarding individual reporting in the primary studies, so we could not construct a 2 × 2 table. In some cases, the lack of appropriate reporting made us judge the study as having an unclear [21, 28, 30, 33, 34] or high risk of bias [27, 31].

The availability of commercial tests demonstrated in recent studies [23, 26] may facilitate the incorporation of the ELISA test into clinical practice, allowing standardized use for the diagnosis of CPA and replacing the reference test that still depends on in-house performance.

Since the global CPA burden is substantial, mainly as a complication of pulmonary tuberculosis (PTB) [38] and especially in countries such as Brazil, which is among the 30 countries representing over 80% of tuberculosis cases worldwide in 2015 [39], there is still a need for well-designed studies to obtain evidence and demonstrate the use of the ELISA tests compared to precipitation tests.

Although it is not possible to define the evidence strength, the clinical implications of this study were as follows: precipitin detection is laborious, requiring specialized laboratories and presenting low sensitivity for the diagnosis of CPA; in-house ELISA tests do not present standard concentrations and antigens for comparative studies; commercial ELISA tests show better performance for diagnosing CPA, but additional studies must be conducted to identify the best cut-off value; and the ImmunoCAP and Immulite systems demonstrated the best performances among commercial tests.

In conclusion, our meta-analysis suggests that the ELISA test presented better accuracy than the precipitation tests (DID and/or CIE) for CPA diagnosis. Thus, ELISA can be considered as the test of choice in clinical practice.

## Supporting information

**S1 Checklist. Prisma 2009.**
(TIF)

**S2 Checklist.**
(TIF)

**S1 Table. Characterization of the studies included in this systematic review and meta-analysis.** ELISA: Enzyme-linked immunosorbent assay; AF: *Aspergillus fumigatus*; Ag: antigen; DID: double immunodiffusion; CPA: chronic pulmonary aspergillosis; OD: optical density; CIE: counterimmunoelectrophoresis; TMB: 3,3′,5,5′-tetramethylbenzidine; pNPP: alkaline phosphatase yellow; OPD: *o*-phenylenediamine; RNU: 18 kDa ribonuclease; DPPV: 88 kDa dipeptidylpeptidase; CAT: 360 kDa catalase.
(TIF)

**S2 Table.**
(TIF)

**S3 Table.**
(TIF)

## Acknowledgments

The authors are grateful for the support of the Coordination for the Improvement of Higher Education Personnel (CAPES), the Foundation for Support to the Development of Education, Science and Technology of the State of Mato Grosso do Sul–FUNDECT and to the National

Council for Scientific and Technological Development–CNPQ for the conduction of this study.

## Author Contributions

**Conceptualization:** Cláudia Elizabeth Volpe Chaves, Sandra Maria do Valle Leone de Oliveira, James Venturini, Antonio Jose Grande, Tatiane Fernanda Sylvestre, Rinaldo Poncio Mendes, Anamaria Mello Miranda Paniago.

**Data curation:** Cláudia Elizabeth Volpe Chaves, Sandra Maria do Valle Leone de Oliveira, James Venturini, Antonio Jose Grande, Tatiane Fernanda Sylvestre, Rinaldo Poncio Mendes, Anamaria Mello Miranda Paniago.

**Formal analysis:** Cláudia Elizabeth Volpe Chaves, Sandra Maria do Valle Leone de Oliveira, James Venturini, Antonio Jose Grande, Tatiane Fernanda Sylvestre, Rinaldo Poncio Mendes, Anamaria Mello Miranda Paniago.

**Investigation:** Cláudia Elizabeth Volpe Chaves, Sandra Maria do Valle Leone de Oliveira, James Venturini, Antonio Jose Grande, Tatiane Fernanda Sylvestre, Rinaldo Poncio Mendes, Anamaria Mello Miranda Paniago.

**Methodology:** Cláudia Elizabeth Volpe Chaves, Sandra Maria do Valle Leone de Oliveira, James Venturini, Antonio Jose Grande, Tatiane Fernanda Sylvestre, Rinaldo Poncio Mendes, Anamaria Mello Miranda Paniago.

**Project administration:** Cláudia Elizabeth Volpe Chaves, Sandra Maria do Valle Leone de Oliveira, James Venturini, Antonio Jose Grande, Tatiane Fernanda Sylvestre, Rinaldo Poncio Mendes, Anamaria Mello Miranda Paniago.

**Resources:** Cláudia Elizabeth Volpe Chaves, Sandra Maria do Valle Leone de Oliveira, James Venturini, Antonio Jose Grande, Tatiane Fernanda Sylvestre, Rinaldo Poncio Mendes, Anamaria Mello Miranda Paniago.

**Software:** Cláudia Elizabeth Volpe Chaves, Sandra Maria do Valle Leone de Oliveira, James Venturini, Antonio Jose Grande, Tatiane Fernanda Sylvestre, Rinaldo Poncio Mendes, Anamaria Mello Miranda Paniago.

**Supervision:** Cláudia Elizabeth Volpe Chaves, Sandra Maria do Valle Leone de Oliveira, James Venturini, Antonio Jose Grande, Rinaldo Poncio Mendes, Anamaria Mello Miranda Paniago.

**Validation:** Cláudia Elizabeth Volpe Chaves, Sandra Maria do Valle Leone de Oliveira, James Venturini, Antonio Jose Grande, Rinaldo Poncio Mendes, Anamaria Mello Miranda Paniago.

**Visualization:** Cláudia Elizabeth Volpe Chaves, Sandra Maria do Valle Leone de Oliveira, James Venturini, Antonio Jose Grande, Rinaldo Poncio Mendes, Anamaria Mello Miranda Paniago.

**Writing – original draft:** Cláudia Elizabeth Volpe Chaves, Sandra Maria do Valle Leone de Oliveira, James Venturini, Antonio Jose Grande, Rinaldo Poncio Mendes, Anamaria Mello Miranda Paniago.

**Writing – review & editing:** Cláudia Elizabeth Volpe Chaves, Sandra Maria do Valle Leone de Oliveira, James Venturini, Antonio Jose Grande, Rinaldo Poncio Mendes, Anamaria Mello Miranda Paniago.

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
