## [Decision Letter · Decision Letter 0]

15 Nov 2019

PONE-D-19-24582

Accuracy of serological tests for diagnosis of chronic pulmonary aspergillosis: a systematic review and meta-analysis

PLOS ONE

Dear Miss VOLPE CHAVES,

Thank you for submitting your manuscript to PLOS ONE. After careful consideration, we feel that it has merit but does not fully meet PLOS ONE’s publication criteria as it currently stands. Therefore, we invite you to submit a revised version of the manuscript that addresses the points raised during the review process.

We would appreciate receiving your revised manuscript by Dec 30 2019 11:59PM. To enhance the reproducibility of your results, we recommend that if applicable you deposit your laboratory protocols in protocols.io, where a protocol can be assigned its own identifier (DOI) such that it can be cited independently in the future. For instructions see: http://journals.plos.org/plosone/s/submission-guidelines#loc-laboratory-protocols

We look forward to receiving your revised manuscript.

Kind regards,

Ritesh Agarwal

Academic Editor

PLOS ONE

Journal Requirements:

-

Reviewers' comments:

Reviewer's Responses to Questions

**Comments to the Author**

1. Is the manuscript technically sound, and do the data support the conclusions?

Reviewer #1: Yes

Reviewer #2: Partly

2. Has the statistical analysis been performed appropriately and rigorously? 

Reviewer #1: Yes

Reviewer #2: N/A

3. Have the authors made all data underlying the findings in their manuscript fully available?

Reviewer #1: Yes

Reviewer #2: No

4. Is the manuscript presented in an intelligible fashion and written in standard English?

Reviewer #1: Yes

Reviewer #2: No

5. Review Comments to the Author

Reviewer #1: Comments for PONE-D-19-24582

Methods: Line 165: The inclusion criteria enlists healthy controls. It could be just written as control group. Sehgal IS have included diseased controls and by this inclusion the study by Sehgal IS et al will be excluded from the review. Generally, diseased controls form a better control group for evaluation of a diagnostic test.

The authors could include the study by Hunter et al (J Clin Microbiol. 2019 Aug 26;57(9). pii: e00538-19. doi: 10.1128/JCM.00538-19). This study also provides information about ImmunoCap Aspergillus IgG and precipitins in subjects with CPA

Could you please discuss about the cut-off values of A.fumigatus specific IgG used in different studies?

Could you provide a best cut-off for A.fumigatus-specific IgG value based on your systematic review?

Please provide reference numbers in the table 1

Please add that diseased controls were used in the study by Sehgal IS, et al. Ref no 27

Line 351: please correct the spelling of sensitivity

Discussion: Lines 401-403: This was not the focus of the current systematic review and may be omitted. Please discuss the short comings of using precipitin tests in discussion. For example these tests are only available at few centers thus are not available to be used for routine practice, even in developing countries. There is lack of standardization of aspergillus antigens that are used across centers. The results of precipitins are semiquantitative and thus do not provide much information

Could you discuss the reason for different cut-off values across different studies? Does it depend on the type of control population used, underlying disease or category of CPA or the ethnicity? For this also use reference Sehgal IS, et al. Efficiency of A. fumigatus-specific IgG and galactomannan testing in the diagnosis of simple aspergilloma. Mycoses. 2019 Aug 13. doi: 10.1111/myc.12987

What are the clinical implications of the current systematic review?

Figure 3: Please arrange studies according to the year of publication for better symmetry

Figures: The quality of all the figures needs to be improved as they appear to be smudged and are not clear

Reviewer #2: Comments:

- The language of the whole manuscript has to be improved by an english native speaker! The first sentence of the abstract (as an example) is no english and hard to read.

- Make two sentences out of the first sentence of the introduction.

- Site 4, line 54: write only CT scan and delete tomographic images

- Site 4, line 56: I would not cite reference 5 in this context. Please delete it.

- Site 4, line 60: it is not a respiratory tree - please write respiratory tract - correction by an native english speaker is urgently needed.

- Site 5, line 85: The epidemiology of A. fumigatus is very diverse and differs by region and continents. I would not write it is 40%, because in Europe it is much higher while in Asia it might be lower. So please report the literature correctly.

- To be honest it only makes sence to continue the review process after the manuscriot has been improved by an english native speaker otherwise it too hard to read.

6. PLOS authors have the option to publish the peer review history of their article (what does this mean?). If published, this will include your full peer review and any attached files.

Reviewer #1: Yes: Inderpaul Singh Sehgal

Reviewer #2: No

---

## [Author Response · Author response to Decision Letter 0]

23 Dec 2019

COMMENTS TO AUTHOR: PONE-D-19-24582

Reviewer #1: 

Methods: 

1) Line 165: The inclusion criteria enlists healthy controls. It could be just written as control group. Sehgal IS have included diseased controls and by this inclusion the study by Sehgal IS et al will be excluded from the review. Generally, diseased controls form a better control group for evaluation of a diagnostic test.

Answer: We changed the inclusion criteria for the control group as suggested in lines 165, 269, and 276 (Table 1). For this reason, we added a subgroup analysis for heterogeneity assessment using only studies with healthy controls in line 171. Results of the subgroup analysis with healthy controls were included in lines 326–329. Figures 7 and 8 were also included for the income statement. We presented the figures in a new sequence for this reason.

2) The authors could include the study by Hunter et al (J Clin Microbiol. 2019 Aug 26;57(9). pii: e00538-19. doi: 10.1128/JCM.00538-19). This study also provides information about ImmunoCap Aspergillus IgG and precipitins in subjects with CPA.

Answer: We welcome the suggestion to include the study by Hunter et al. We included this study for qualitative evaluation as suggested.

3) Could you please discuss about the cut-off values of A.fumigatus specific IgG used in different studies?

Answer: We included the discussion of cut-off values in lines 385–389 and the article reference (35) as suggested. We included a column with the cut-off values for the ELISA tests studies evaluated in the review in Table 1.

4) Could you provide a best cut-off for A.fumigatus-specific IgG value based on your systematic review?

Answer: Based on individual tests, we included the best ImmunoCAP system cut-off in lines 282–283 and 383–384, as suggested. 

5) Please provide reference numbers in the Table 1

Answer: We included the reference numbers in Table 1 and changed the sequence of references throughout the study because we removed two studies and included two other studies based on the suggestions. We also included the reference numbers and made some additional revisions in S1 Table.

6) Please add that diseased controls were used in the study by Sehgal IS, et al. Ref no 27

Answer: We included the use of diseased controls for the study by Sehgal IS et al., lines 212 and 276 (Table 1), as suggested. Given the revisions to the references, we changed the citation number of Sehgal IS et al to [26].

7) Line 351: please correct the spelling of sensitivity

Answer: We corrected the spelling of the word “sensitivity” in line 341. 

Discussion: 

8) Lines 401-403: This was not the focus of the current systematic review and may be omitted. 

Answer: We eliminated the discussion in lines 401–403.

9) Please discuss the short comings of using precipitin tests in discussion. For example these tests are only available at few centers thus are not available to be used for routine practice, even in developing countries. There is lack of standardization of aspergillus antigens that are used across centers. The results of precipitins are semiquantitative and thus do not provide much information.

Answer: We discussed the limitation of precipitin tests in lines 393–396, as suggested, and we have included reference 36.

10) Could you discuss the reason for different cut-off values across different studies? Does it depend on the type of control population used, underlying disease or category of CPA or the ethnicity? For this also use reference Sehgal IS, et al. Efficiency of A. fumigatus-specific IgG and galactomannan testing in the diagnosis of simple aspergilloma. Mycoses. 2019 Aug 13. doi: 10.1111/myc.12987

Answer: We included the discussion of cut-off values in lines 385–389 and the article reference (35), as suggested.

11) What are the clinical implications of the current systematic review?

Answer: We have now included the clinical implications of the study in lines 418–424. 

12) Figure 3: Please arrange studies according to the year of publication for better symmetry

Answer: Unfortunately, we cannot arrange studies by year of publication because the Review Manager tool describes the studies in alphabetical order.

13) Figures: The quality of all the figures needs to be improved as they appear to be smudged and are not clear.

Answer: We have produced new figures with better quality for the revised submission.

Reviewer #2: 

14) The language of the whole manuscript has to be improved by an english native speaker! The first sentence of the abstract (as an example) is no english and hard to read.

Answer: We welcome your suggestion and we will communicate with Editage about the quality of translation into English. We have supplemented the initial translation certificate and the new resubmission certificate. Lines 22–24 of the abstract have now been revised.

15) - Make two sentences out of the first sentence of the introduction.

Answer: We changed the two sentences as suggested. 

16) - Site 4, line 54: write only CT scan and delete tomographic images

Answer: We have deleted the words “tomographic images” in line 55 as suggested.

17) - Site 4, line 56: I would not cite reference 5 in this context. Please delete it.

Answer: We deleted reference 5 as suggested and re-arranged the remaining references. 

18) - Site 4, line 60: it is not a respiratory tree - please write respiratory tract - correction by an native english speaker is urgently needed.

Answer: We replaced the words “respiratory tree” with “respiratory tract” in line 61.

19) - Site 5, line 85: The epidemiology of A. fumigatus is very diverse and differs by region and continents. I would not write it is 40%, because in Europe it is much higher while in Asia it might be lower. So please report the literature correctly.

Answer: We corrected the description in lines 85–87.

20) - To be honest it only makes sence to continue the review process after the manuscript has been improved by an english native speaker otherwise it too hard to read.

Answer: We welcome your suggestion, and we will communicate with Editage about the quality of translation into English. We have supplemented the initial translation certificate and the new resubmission certificate. .

Note: The sentence from lines 77 to 80 has been revised per the request of Dr. van Toorenemberg cited as reference 7 (previous) and current reference 6, by pre-print edition. The previous sentence was “Historically, IgG ELISA assays used in-house antigens, with different antigenic preparations and concentrations, which makes the comparison of test performance very difficult [7].”

Sincerely yours, 

Cláudia Elizabeth Volpe Chaves, Msc. Corresponding author.

Phone: +55-67-33782701; Av. Engenheiro Luthero Lopes, n.36 – Bairro Aero Rancho V, Campo Grande, MS, Brazil, CEP 79.084-180. E-mail: claudiavolpe70@hotmail.com (C.E. Volpe-Chaves).

---

## [Decision Letter · Decision Letter 1]

23 Jan 2020

Accuracy of serological tests for diagnosis of chronic pulmonary aspergillosis: a systematic review and meta-analysis

PONE-D-19-24582R1

Dear Dr. VOLPE CHAVES,

We are pleased to inform you that your manuscript has been judged scientifically suitable for publication and will be formally accepted for publication once it complies with all outstanding technical requirements.

With kind regards,

Ritesh Agarwal

Academic Editor

PLOS ONE

Additional Editor Comments (optional):

-

Reviewers' comments:

Reviewer's Responses to Questions

**Comments to the Author**

1. If the authors have adequately addressed your comments raised in a previous round of review and you feel that this manuscript is now acceptable for publication, you may indicate that here to bypass the “Comments to the Author” section, enter your conflict of interest statement in the “Confidential to Editor” section, and submit your "Accept" recommendation.

Reviewer #1: All comments have been addressed

2. Is the manuscript technically sound, and do the data support the conclusions?

Reviewer #1: Yes

3. Has the statistical analysis been performed appropriately and rigorously? 

Reviewer #1: Yes

4. Have the authors made all data underlying the findings in their manuscript fully available?

Reviewer #1: Yes

5. Is the manuscript presented in an intelligible fashion and written in standard English?

Reviewer #1: Yes

6. Review Comments to the Author

Reviewer #1: (No Response)

7. PLOS authors have the option to publish the peer review history of their article (what does this mean?). If published, this will include your full peer review and any attached files.

Reviewer #1: Yes: Inderpaul Singh Sehgal

---

## [Editor Report · Acceptance letter]

28 Jan 2020

PONE-D-19-24582R1 

Accuracy of serological tests for diagnosis of chronic pulmonary aspergillosis: a systematic review and meta-analysis 

Dear Dr. VOLPE CHAVES:

I am pleased to inform you that your manuscript has been deemed suitable for publication in PLOS ONE. Congratulations! Your manuscript is now with our production department. 

With kind regards,

on behalf of

Dr. Ritesh Agarwal 

Academic Editor

PLOS ONE